# General object-based features account for letter perception

**Daniel Janini** [1] *, **Chris Hamblin** [1], **Arturo Deza** [1,2], **Talia Konkle** [1]

**1** Department of Psychology, Harvard University, Cambridge, Massachusetts, United States of America,
**2** Department of Brain and Cognitive Sciences, Massachusetts Institute of Technology, Cambridge, Massachusetts, United States of America

* daniel_janini@g.harvard.edu

**Data Availability Statement:** The behavioral data from both experiments and the code to reproduce the analyses of this study are available at: https://osf.io/8s3vy/. This repository includes data at the trial level for each participant and the stimuli used in the experiments. Code is included to construct

## Abstract

After years of experience, humans become experts at perceiving letters. Is this visual capacity attained by learning specialized letter features, or by reusing general visual features previously learned in service of object categorization? To explore this question, we first measured the perceptual similarity of letters in two behavioral tasks, visual search and letter categorization. Then, we trained deep convolutional neural networks on either 26-way letter categorization or 1000-way object categorization, as a way to operationalize possible specialized letter features and general object-based features, respectively. We found that the general object-based features more robustly correlated with the perceptual similarity of letters. We then operationalized additional forms of experience-dependent letter specialization by altering object-trained networks with varied forms of letter training; however, none of these forms of letter specialization improved the match to human behavior. Thus, our findings reveal that it is not necessary to appeal to specialized letter representations to account for perceptual similarity of letters. Instead, we argue that it is more likely that the perception of letters depends on domain-general visual features.

## Author summary

For over a century, scientists have conducted behavioral experiments to investigate how the visual system recognizes letters, but it has proven difficult to propose a model of the feature space underlying this capacity. Here we leveraged recent advances in machine learning to model a wide variety of features ranging from specialized letter features to general object-based features. Across two large-scale behavioral experiments we find that general object-based features account well for letter perception, and that adding letter specialization did not improve the correspondence to human behavior. It is plausible that the ability to recognize letters largely relies on general visual features unaltered by letter learning.

RDMs from the behavioral data, construct RDMs from the neural network models, conduct model-behavior correlations and statistical tests, and plot the results in figures.

**Funding:** This work was supported by NSF CAREER BCS-1942438 (T.K.), and the National Defense Science and Engineering Graduate Fellowship Program (D.J.). URLs of funders: https://beta.nsf.gov/funding/opportunities/faculty-early-career-development-program-career https://ndseg.sysplus.com/ The funders played no role in study design, data collection and analysis, decision to publish, or preparation of the manuscript.

**Competing interests:** The authors have declared that no competing interests exist.

## Introduction

A hallmark achievement of the human visual system is its ability to distinguish between thousands of objects. It has long been theorized that this capacity is achieved with a rich multidimensional feature space, where different features of the space detect the presence of different shape and texture properties [1–4]. Another major achievement of the human visual system is its ability to distinguish between stimuli within specific domains (e.g., faces, letters). Long-standing debates concern which domains are represented with specialized mechanisms separate from the general mechanisms supporting object recognition [5–11]. Letters are one domain of visual stimulus that literate humans expertly perceive. For example, if you are reading this paper visually, you have rapidly and effortlessly perceived over 600 letters since the start of this paragraph. What feature space allows for letters to be so easily perceived–the same feature space that supports object perception, or a feature space learned specifically for letters?

Functional neuroimaging studies indicate that learning to read alters the functional responses of visual cortex, but these findings have left open the nature of the visual feature space that underlies our perception of letters. It is clear that the intensive process of learning to read reshapes the macro-scale organization of visual cortex [5,12,13]. Most prominently, in literate adults, a region of inferotemporal cortex termed the visual word form area exhibits preferential responses to letter strings, with perhaps the highest responses reserved for the letters of one's own alphabet [14–17]. However, these brain-based changes are compatible with multiple learning stories, leaving open competing possibilities for the nature of features underlying letter perception. One possibility is that the visual system learns the specific visual features that are ideally tuned for categorizing the letters of one's alphabet, which we will refer to as "specialized letter features". Given the extensive reading instruction children receive and how often humans read in daily life, one might expect that the visual system learns specialized letter features. Alternatively, the visual system may learn how to categorize letters by reusing more general visual features which can support object representation. This latter possibility is articulated in the "neuronal recycling hypothesis" [12,18], where a recycling process could leave the object-based features unaltered, or it could alter the object-based features with some degree of experience-dependent specialization.

Over a century of behavioral experiments have established methods for measuring the representational similarity between visually presented letters, giving researchers a window into the underlying feature space [19–24]; historically, however, researchers have struggled to articulate a set of visual features that can account for these data [25]. Similarity measurements are thought to reflect the feature space in which stimuli are represented, with perceptually similar stimuli sharing similar patterns of feature activation [26]. Some early work attempted to use data-driven approaches to infer the feature space in which letters are represented, applying dimensionality reduction techniques to similarity measurements [1,20,27]. Hypothesis-driven approaches have also been employed, positing that letters are represented by interpretable features such as line orientations, curves, terminations, and intersections [1,28–30]. Throughout all this research, the features considered have been relatively simple visual properties accessible to experimenter intuition, and it has been nearly impossible for researchers to consider more complex object-based feature spaces. Indeed, as early as the 1960s, researchers studying letter perception lamented their dependence on intuition and wished for models capable of learning feature spaces on their own [1,3].

In the past decade, vision scientists have gained access to such models with the advent of deep convolutional neural networks, which are capable of learning features for a variety of tasks [31,32]. In particular, neural networks trained to categorize objects have shown promise for capturing feature spaces that are similar to those of the human visual system [33],

predicting neural responses to objects [34–39] and behavioral measurements of object similarity [40,41]. Further, these networks are not trained to match the human visual system, so their correspondence to human perception is a natural consequence of the structure of the natural images, the constraints of the trained task (e.g., categorization), and the prior of the deep convolutional network architecture. Most importantly for the sake of this paper, the nature of the feature spaces learned by a neural network are under experimental control. A convolutional neural network trained to categorize letter images into one of twenty-six categories will learn a hierarchy of feature spaces that are specialized for letter categorization, as the model only ever received letter images as input and only ever learns features to support letter classification. The same network architecture trained to categorize object images into one of a thousand categories will instead learn a hierarchy of more general object-based feature spaces. In this way, we used convolutional neural networks to operationalize different kinds of feature spaces, enabling us to explore whether these domain-specialized or domain-general features better account for the perception of letters.

In this project, we investigated the visual representations supporting letter perception by taking a behavioral-computational approach. We first measured the similarity of Roman alphabet letters in two behavioral tasks, visual search and letter categorization. Then, through the use of representational similarity analysis [42], we compared how well the features spaces learned from letter-trained vs object-trained deep convolutional neural networks could account for the behaviorally measured similarity of letters (**Fig 1A**). To preview the main results, we found that the object-trained features showed a more consistent correspondence to the behavioral tasks. Further, different attempts to alter these general-object-based features with experience-dependent letter specialization did not improve their ability to account for the behavioral data. These findings lend computational plausibility to the theoretical position that letter perception is supported by general object-based features. By extension, we hypothesize that intensive letter learning may reflect the construction of read-out mechanisms which access (rather than alter) this general visual feature space.

## Results

### Letter search

We measured the perceptual similarity of letters in a large-scale online visual search task. Specifically, participants (n = 222) located the odd-letter-out as quickly as possible in displays with one letter among five other letters of a different identity (e.g., the letter a among five letter b's; **Fig 1B**). Visual search is faster when the target is more perceptually distinct from the distractors, and it is slower when the target is more similar to the distractors [43]; in this way, visual search time serves as an implicit measure of the perceptual similarity between stimuli [44–49]. Reaction times were measured for all pairs of 26 lowercase letters in 20 fonts, yielding a 26x26 representational dissimilarity matrix (RDM) reflecting the pairwise similarities of letters averaged across font variation. This experimental paradigm yielded a highly reliable RDM across participants and fonts (Spearman-Brown Corrected Reliability ρ = 0.89, see Methods). The perceptual similarity of letters is visualized with a multi-dimensional scaling projection in **Fig 1B**.

Next, we leveraged convolutional neural networks to operationalize different features. To operationalize general object-based features, we considered the features of AlexNet trained to do 1000-way object classification on the ImageNet database [31,50]. Even though this object-trained network was never trained to distinguish between letter identities, its features can generalize to the task of letter recognition by supporting the linear decoding of letter identity across font and size variation (mean pairwise decoding accuracy of 95–97% in Layers 3–7, see

## A. Neural Network Model Feature Spaces

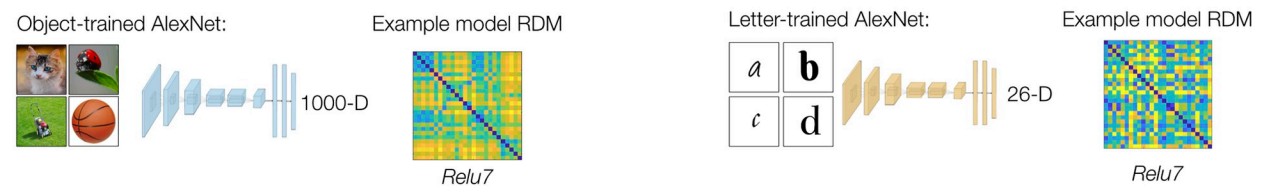

## B. Experiment 1: Visual Search

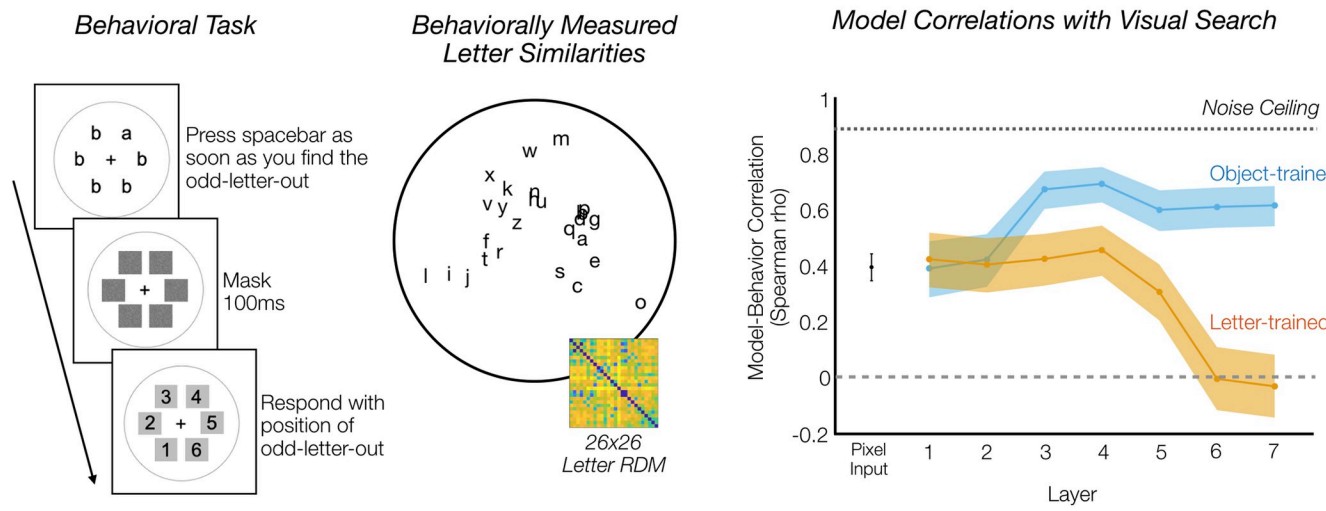

## C. Experiment 2: Letter Categorization

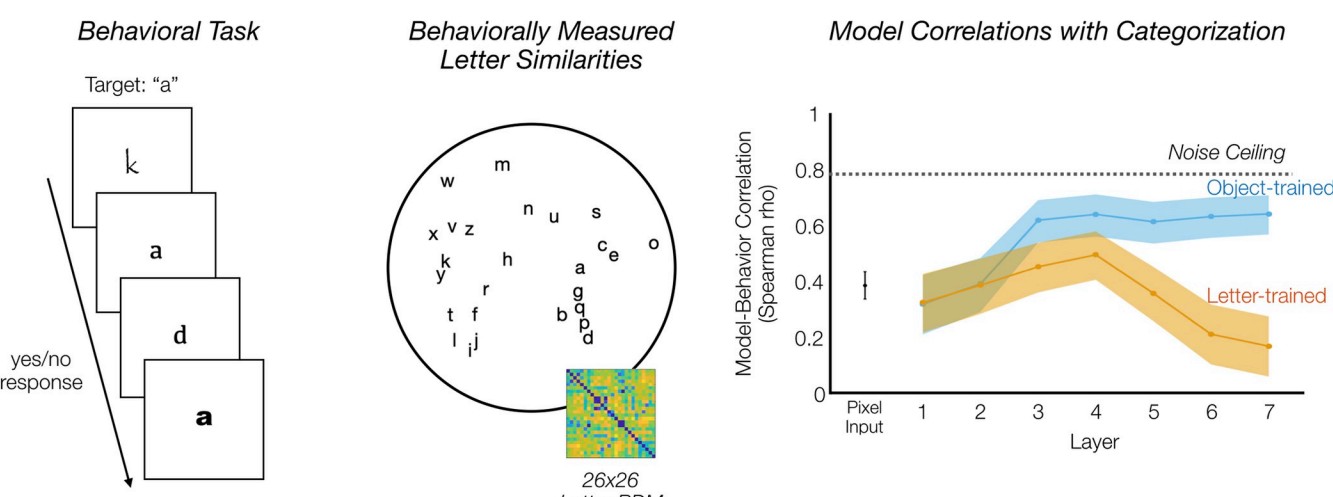

**Fig 1. Results from the Visual Search and Categorization Experiments.** A. Neural networks used to operationalize general object features and specialized letter features. AlexNet trained on 1000-way object classification was used as a model of general object-based features, and AlexNet trained on 26-way letter classification was used as a model of specialized letter features. B. Experiment 1: Visual Search. The structure of the behavioral task is displayed on the left. Reaction time was considered a measure of the perceptual similarity of the two letters, with slower reaction times indicating greater similarity. In the middle is a visualization of the pairwise letter similarities measured in the visual search task. More similar letters are closer together in this multidimensional scaling display. On the right, model-behavior correlations between the model RDMs and the RDM from the visual search experiment are plotted on the y-axis, as a function of the layer of the AlexNet model. The shaded error range indicates the 95% confidence interval across bootstrapped samples of letter pairs. C. Experiment 2: Letter Categorization. Example trials are shown on the left. Participants were given a target letter (e.g., "a") and categorized each letter as "a" or "not a" as quickly as possible. The time to reject a letter was a measure of its similarity to the target letter. In the middle is a visualization of the pairwise letter

similarities measured in the categorization experiment. On the right, model-behavior correlations are plotted as before. Cat: https://commons.wikimedia.org/wiki/File:Tabby_cat_with_blue_eyes-3336579.jpg. Ladybug: https://commons.wikimedia.org/wiki/File:Ladybird.jpg. Lawnmower: https://commons.wikimedia.org/wiki/File:Tondeuse.png. Basketball: https://commons.wikimedia.org/wiki/File:Basketball.jpeg.

S1 Fig). To operationalize specialized, domain-specific features, we trained another AlexNet model only on letter images, tasked to do letter classification across font variability and a variety of augmentations, over both typeset and handwritten characters (see Methods). The letter-trained model achieved 95.2% Top-1 accuracy on letter classification (S1 Table).

First, we examined the extent to which these two networks differed in how they represent letter information. To measure the representational structure of letters learned in these two different networks, we constructed model RDMs for each layer by measuring the feature activations to each letter stimulus and calculating the pairwise distances between letters in these feature spaces. This procedure yielded a hierarchical set of object-trained and letter-trained model RDMs. Next, we computed the correlation between the object-trained RDM and the letter-trained RDM for each layer. Correlations between the two models' RDMs were high in early layers (r = 0.98 and r = 0.88 for Layers 1 and 2) but decreased by the later layers (r = 0.36 and r = 0.30 for Layers 6 and 7; see full list of correlations in S2 Fig). Finally, we also confirmed that the letter-trained network did not drive toward a purely identity-based representation without graded levels of similarity between letters. The later layers of both the object-trained and letter-trained networks showed only weak correlations with a purely categorical representational space (letter-trained network: r = 0.30, r = 0.26; object-trained network: r = 0.28, r = 0.27, for Layers 6 and 7, respectively). Thus, both the object-trained and letter-trained networks have feature spaces that can support successful letter classification, but they differ in terms of the feature spaces that support this performance, yielding different predictions about the graded similarity relationships among the letters.

How well do the feature spaces from the object-trained network and letter-trained network correspond to the representational space measured when human participants distinguish between letters? To answer this question, we calculated the Spearman correlation between each of the model RDMs and the perceptual RDM measured in the visual search task. The results are plotted in **Fig 1B**. Across the layers, the object-trained feature spaces yielded model-behavior correlations that were moderate in early layers and stronger in the mid-to-late layers. The letter-trained feature spaces also yielded moderate model-behavior correlations in Layers 1–4, but model-behavior correlations were weak to nonexistent in Layers 5–7. Direct statistical comparisons between these models at each layer showed that the object-trained network exhibited higher model-behavior correlations in Layers 3–7 ($p < 10^{-4}$ for all layers, bootstrap resampling of letter pairs), while the letter-trained network exhibited a higher model-behavior correlation only in Layer 1 ($p = 7.2^{*}10^{-4}$, bootstrap resampling of letter pairs). In addition, we computed model-behavior correlations separately for each font tested, and we found that the maximum model-behavior correlations were higher for the object-trained network than the letter-trained network ($t(19) = 14.07$, $p < 10^{-4}$).

These results show that at least one layer of both the letter-trained network and object-trained network exhibited a decent correspondence with the perceptual similarity of letters. However, when comparing the two models, the object-trained model exhibited more consistent model-behavior correlations in the mid-to-late layers.

## Letter categorization

While our visual search experiment required participants to discriminate between letters, we note that visual search does not require participants to perform explicit letter categorization.

For example, it is possible to complete visual search with letters from an alphabet one cannot read. Thus, we next conducted a second large-scale behavioral experiment in which we measured the perceptual similarity of letters using a categorization task. Both visual search and categorization are perceptual tasks, but the two tasks require different processes. Categorization involves holding a target letter template in mind and comparing it to the incoming stimulus, while visual search involves distinguishing between simultaneously presented stimuli. Thus, it is possible that the categorization task will measure a distinct representational structure of letters more suited to explicit categorization. This representational structure may better correspond with the letter-trained network given that this network was trained to categorize letters. On the other hand, prior empirical evidence shows that visual search speeds and categorization speeds are deeply related and may actually be constrained by the same underlying perceptual representational bottleneck [46,51].

In the categorization experiment, single letters were presented at fixation and the task was to categorize each letter as quickly as possible. Participants (n = 517) were given a target letter (e.g., "a") and responded whether each presented letter was the target or not ("a" or "not a"; **Fig 1C**). The critical trials were those in which the presented letter was *not* the target. We measured the time it took to reject these letters as an index of their similarity to the internally represented target letter (see also Cohen et al., 2017 for this method). By assigning each letter as the target across sets of trials, we measured the perceptual similarity of each possible letter pairing. Each participant could only be tested on a subset of letter pairings in a reasonable experiment duration, so we systematically distributed letter pairings across participants and used linear mixed effect modeling to estimate the full 26x26 letter RDM (see Methods). This experiment and modeling procedure yielded a reliable RDM across participants (Spearman-Brown Corrected Reliability $\rho$ = 0.78, see Methods). The mean accuracy among included participants was 97.2±2.0%.

Next, we tested our main question: how well do general object-based features and specialized letter features account for letter similarity measured during categorization? The correlations between the behaviorally measured RDM and the model RDMs from object-trained AlexNet and letter-trained AlexNet are shown in **Fig 1C**. The object-trained network showed the highest correlation with the categorization-based RDM ($p$ = 0.00044, bootstrap resampling of letter pairs), with lower model-behavior correlations in early layers and higher correlations in mid-to-late layers. Model-behavior correlations from the letter-trained network did increase some from low-to-mid-layers but not to the same extent as the object-trained network, and correlations decreased in later layers. When comparing each layer between the two networks, the object-trained network exhibited higher model-behavior correlations in Layers 3–7 ($p < =$ 0.00076 in each layer), and the two networks did not differ in Layers 1 and 2 ($p$ = 0.6266 and $p$ = 0.8732, respectively).

Thus, the results from the categorization experiment were like those found in the visual search experiment. Specialized letter features were not necessary to account for human letter perception, as object-based feature spaces best accounted for the perceptual similarity of letters, with features of mid-to-late layers showing the highest correspondence.

## Comparisons between visual search and categorization tasks

Direct comparisons of the two behavioral experiments reveal similar representational structure. That is, there was a high correlation between the RDMs measured in our visual search and categorization experiments ($\rho$ = 0.71). In fact, the correlation between the two experiments was nearly as high as the noise ceiling of the categorization-based RDM ($\rho$ = 0.78), implying that almost all the reliable variance in the categorization-based RDM was accounted

for by the visual search RDM. In contrast, the correlation between the two experiments was a bit lower than the noise ceiling of the visual search RDM ($\rho = 0.89$). MDS plots depicting the representational space of letters from the two experiments, in addition to the representational spaces from object-trained and letter-trained AlexNet, can be viewed in S2 Fig. Thus, both experiments largely measured the same representational space for letters, but the RDM from the visual search experiment contained some reliable variance not measured in the categorization experiment. Our data thus indicate that there is a similar representational structure underlying both perceptual tasks that is accounted well by general object-based features.

## Models of object-based features with experience-dependent specialization

So far, we have investigated two extremes in a spectrum of possibilities: from purely letter-specialized features on one end, to object-based features unaltered by experience with letters on the other end. However, a variety of feature spaces exist between these two extremes involving object-based features that are subsequently altered by letter learning. Further, while close, no models have yet reached the noise ceiling of the behavioral data, so there is some reliable, behaviorally relevant representational structure not accounted for by the object-trained model RDMs. Thus, we next altered object-based spaces with letter specialization, using three different approaches which each operationalize a different hypothesis about how experience-dependent specialization could be accomplished.

First, we considered fine-tuning operations. Perhaps extensive visual practice with letters alters the tuning of object-based features learned prior to letter training. To explore this possibility, we created a fine-tuned model in which AlexNet was first trained on object classification, then next trained to categorize both objects and letters, with the final 1000-way object classification layer replaced by a 1026-way classifier for 1000 objects and the 26 letters. By fine-tuning with a mixture of objects and letters, we created a network that could classify both objects and letters (see S1 Table), preventing the "catastrophic forgetting" of previously learned features that can occur when training a network on a sequence of tasks [52]. Comparing the original and fine-tuned networks, we found that the learned feature spaces were similar through the early and mid-layers (RDM correlations across layers 1–5: range: $r = 0.795$–$0.995$), while the RDMs from the fully connected layers started to show more divergence ($r = 0.68$ and $r = 0.57$ in Layers 6 and 7, respectively), opening the possibility that the fine-tuned model would better account for the perceptual similarity of letters. However, as shown in **Fig 2A**, the highest model-behavior correlation in the fine-tuned network was either worse than in the original object-trained network ($p = 0.0037$ for visual search) or they did not differ ($p = 0.9980$ for letter categorization, bootstrap resampling of letter pairs).

Additionally, we created another fine-tuned model in which letter training alone followed object training, with the final 1000-way object classification layer replaced by a 26-way letter classifier. Again, this method of fine-tuning either did not significantly affect the maximum model-behavior correlation ($p = 0.0848$ for visual search) or actually decreased the network's maximum model-behavior correlation ($p = 0.00024$ for letter categorization). Thus, fine-tuning object-based features with letter training did not yield representations more similar to the representational structure evident in behavior.

Next, we considered branching networks. Perhaps specialized letter features are built from object-based features by branching at some stage of the object-based hierarchy. To explore this account, we created a family of branching neural networks–each a five-layer neural network with input from one of the first five layers of object-trained AlexNet. The branches were trained to perform letter classification without altering the object-based features they took as input (see Methods). Results are shown in **Fig 2B**. None of these branches exhibited a higher

## Model-Behavior Correlations

### A. Finetuning

### B. Network Branches

### C. Letter-preferring subspace

**Fig 2. Experience-dependent specializations for letters.** A. Fine-tuning. An object-trained network was fine-tuned on letters alone (yellow) or with objects and letters (red). Layer-wise model-behavior correlations are shown for visual search (center), and letter categorization (right). The dashed-blue line indicates the performance of an object-trained network, for reference. B. Branching networks. Five branching networks were trained, with the input to each network from each of the first five ReLU layers of AlexNet trained on ImageNet and trained to do 26-way letter categorization (dark red to light orange). Adjacent plots show the model-behavior correlation of these networks, beginning with the final object-trained layer (dashed lines), followed by the specialized hierarchical feature spaces learned in the 5-layer branching networks. C. Subspace. In each layer of object-trained AlexNet, features were identified that showed higher activation to letter stimuli than objects, and the representational space for letters was measured in this letter-preferring

subspace. The model-behavior correlations are shown in adjacent subplots. The shaded error range indicate the 95% confidence interval across bootstrapped samples of letter pairs. Note that these shaded error ranges have been omitted for (A) and (B) for visualization clarity but were conducted for statistical tests. Cat: https://commons.wikimedia.org/wiki/File:Tabby_cat_with_blue_eyes-3336579.jpg. Ladybug: https://commons.wikimedia.org/wiki/File:Ladybird.jpg. Lawnmower: https://commons.wikimedia.org/wiki/File:Tondeuse.png. Basketball: https://commons.wikimedia.org/wiki/File:Basketball.jpeg. Bear: https://commons.wikimedia.org/wiki/File:Ursus_arctos_in_Junsele.jpg. Elephant: https://commons.wikimedia.org/wiki/File:Elephas_maximus_(Asiatic_elephant),_Burgers_zoo,_Arnhem,_the_Netherlands.jpg. Microphone: https://commons.wikimedia.org/wiki/File:Microphone_studio.jpg. Pepper: https://commons.wikimedia.org/wiki/File:Red-Pepper.jpg. Pretzel: https://commons.wikimedia.org/wiki/File:Gr%C3%BCndonnerstags-Brezel.jpg.

maximum model-behavior correlation than object-trained AlexNet. Because object-trained AlexNet exhibited high model-behavior correlations in Layers 3–5, one might have expected that branches from these layers would have transformed the feature space to better match the behaviorally measured similarity of letters. In fact, the opposite occurred, and for each of these network branches, model-behavior correlations decreased from the object-trained input space to the final layer of the specialized letter space (each $p < 10^{-4}$ for both visual search and letter categorization, bootstrap resampling of letter pairs). If anything, transitioning the representational space from object-based to letter-based made the resulting structure less like the structure evident in our behavioral experiments.

In a similar vein as the branching networks, we also tested a previously published model from Testolin et al. [53], which learned specialized letter features operating over a bank of general low-level features. We again found that the features of the object-trained AlexNet exhibit higher correlations with both behaviors than the specialized letter features ($p < 10^{-4}$ for both visual search and letter categorization, bootstrap resampling of letter pairs; see S3 Fig).

Finally, we explored the possibility that there is a letter-selective subspace embedded within the object-trained feature space, which might better capture the behaviorally-measured perceptual similarity among letters. In above analyses, the object-trained model RDMs were computed using all the features of each layer of AlexNet; however, one possibility is that letters are specifically represented by those features which are preferentially activated by letters. Thus, we tested whether a subset of the object-trained features exhibit letter-selectivity, even without letter training; and if so, whether these letter-selective features constitute a feature subspace that better accounts for the behavioral data.

To do so, we identified features in object-trained AlexNet which preferentially responded to letters over objects, following procedures in Prince & Konkle [54]. Between 3–22% of features in each layer preferentially responded to letters. We calculated the correlations between the RDMs from these letter-selective features and the RDMs from our visual search and letter categorization experiments. Results are shown in **Fig 2C**. The highest model-behavior correlation from the letter-preferring features did not differ from the highest model-behavior correlation from the full object-based feature spaces ($p = 0.2104$ for visual search, and $p = 0.1323$ for letter categorization, bootstrap resampling of letter pairs), nor did it differ from a matched number of randomly selected non-letter preferring features ($p = 0.1824$ for visual search, and $p = 0.0812$ for letter categorization, bootstrap resampling of letter pairs). We also conducted the same analysis for the letter-preferring features of AlexNet fine-tuned on letters and objects, and the maximum model-behavior correlations were either lower than AlexNet trained on object classification alone ($p = 0.0033$ for visual search) or they did not differ ($p = 0.5494$ for letter categorization, bootstrap resampling of letter pairs). Thus, it is not the case that there is a letter-preferring subspace of the object-based feature space that better captures the perceptual similarity of letters.

Taken together, across these theoretically distinct approaches for how letter-based specialization could modify object-based feature spaces, we did not see any improvements in the

correlations with the behaviorally measured similarity of letters. In fact, when experience-dependent specializations did change the model-behavior correlations, it was for the worse. Thus, the general feature spaces of AlexNet trained on ImageNet were the best of the models we explored, accounting very well for the similarity of letters, though a small amount of reliable variance in the behavioral similarity space of letters remains to be explained.

## Additional model comparisons

We conducted several robustness tests and comparison models to further contextualize the results from object-trained AlexNet and letter-trained AlexNet. First, we compared the object-trained and letter-trained networks to a pixelwise model of similarity. The object-trained network exhibited higher maximum model-behavior correlations than the pixelwise model: $p < 10^{-4}$ for both visual search and categorization), while the letter-trained model did not consistently exhibit higher model-behavior correlations than the pixelwise model ($p = 0.2028$ for visual search and $p = 0.0281$ for categorization, bootstrap resampling of letter pairs). These comparisons indicate that the object-trained network learned humanlike visual representations of letters beyond the low-level image input. Next, we compared the two primary networks to AlexNet with random weights (see S4 Fig). The object-trained network exhibited a higher maximum model-behavior correlation than AlexNet with random weights (object-trained model: $p < 10^{-4}$ for both visual search and categorization), while the letter-trained model did not consistently exhibit higher model-behavior correlations than AlexNet with random weights (p = 0.8961 for visual search and $p = 0.00036$ for categorization, bootstrap resampling of letter pairs). Thus, the neural network architecture alone was not sufficient to create representational structure with a strong match to human behavior, and training on object classification created more humanlike representations.

ImageNet-trained neural networks tend to represent local shape and texture features rather than global contours [55,56], raising the possibility that a model less biased toward texture features would better account for the behavioral data in this study. While creating neural networks with global shape features is still an ongoing endeavor for the field, here we considered another object-trained network trained on stylized ImageNet, which decreases the network's bias toward texture features [56]. For the visual search experiment, the maximum model behavior-correlation from AlexNet trained on stylized ImageNet (ρ = 0.72) was a bit higher than the maximum model-behavior correlation from the typical object-trained network (ρ = 0.69; $p = 0.0338$, bootstrap resampling of letter pairs). However, for the categorization experiment, maximum model-behavior correlations did not significantly differ between stylized-ImageNet-trained AlexNet and the typical ImageNet-trained AlexNet ($p = 0.1575$, bootstrap resampling of letter pairs). These findings indicate that there is a consistent correspondence between object-trained models and human letter perception, though future engineering of object-trained neural networks with global contour representations may improve the ability to account for the behavioral data found in this study.

Previously, we kept the architecture constant between the object- and letter-trained networks to ensure that differences in learned features were due to the input image sets alone. However, while large architectures are needed to solve object categorization, much smaller networks can be trained to accurately classify written symbols [57]. A smaller network with fewer layers and features may learn different representations from a larger network, and it may be less likely to overfit to the training set. Therefore, we tested whether a smaller network would better approximate the behavioral data from our two experiments. We created a smaller architecture which had fewer layers and fewer features per layer than AlexNet, then trained this architecture on letter categorization (see Methods). While this model could also classify

letters accurately, it only exhibited moderate correlations with the perceptual similarity of letters as measured in the visual search and categorization experiments (see S5 Fig). In both experiments, object-trained AlexNet exhibited a higher maximum model-behavior correlation than the small letter-trained network ($p < 10$^-4 for visual search, $p = 0.00036$ for categorization, bootstrap resampling of letter pairs). Thus, a smaller letter-trained network still did not better account for perceptual similarity of letters in comparison to the object-trained model.

Next, we investigated whether training an AlexNet to classify letters that were superimposed on scene backgrounds would create a model that could close the gap with the noise ceiling of our behavioral data. However, layer-wise comparisons showed that this network never exhibited higher model-behavior correlations than object-trained AlexNet. Like the other AlexNet models trained on letters, this network drove toward a representational space with little correspondence to the behaviorally measured similarity of letters (Layer 7 model-behavior correlations: $\rho = 0.02$ and $\rho = 0.20$ for visual search and letter categorization, respectively).

In earlier iterations of this project, our letter-trained networks were trained on only typeset images with a less variable augmentation scheme. In comparison to the main letter-trained networks reported in this study, the previous letter-trained networks showed an even weaker correspondence with the perceptual similarity of letters (see S6 Fig). These model explorations highlight that in addition to the domain of input images and the classification task, the scope of data augmentations has a clear impact on the formation of neural network feature spaces, and their resulting correspondence with perceptual similarity measures.

Finally, we considered a set of intuitive features previously proposed in the literature on letter perception, including line orientations, curves, intersections, and terminations [28–30]. The model RDM computed from these intuitive features exhibited a moderate correlation with the visual search RDM ($\rho = 0.42$ for visual search and $\rho = 0.52$ for categorization); however, object-trained AlexNet again exhibited higher model-behavior correlations ($p < 0.001$ for both visual search and categorization, bootstrap resampling of letter pairs).

## Discussion

Here we probed the nature of the representations supporting letter perception, using a behavioral-computational approach. We trained a set of deep convolutional neural networks to operationalize different feature spaces, ranging from fully letter-specialized to more general and object-based, and estimated how well they could account for the behavioral similarity structure underlying two different letter perception tasks. We found that 1) the general object-based feature spaces best accounted for behavioral data on letter perception in comparison to the specialized letter features; 2) various attempts to add experience-dependent letter representations to the object-based features did not improve how well our models accounted for the behavioral data. Taken together, these results support the plausibility of the claim that letter perception is supported by a set of more general object-based features that can discriminate among many kinds of visual input.

Our findings develop the neuronal recycling hypothesis by comparing multiple ways in which object-based features could be adapted for letter perception. This framework proposes that learning to recognize letter strings involves the "minimal adaptation" of object representations in inferotemporal cortex [58,59]. However, a range of possible mechanisms with varying degrees of letter specialization are compatible with this learning account, from learning linear classification boundaries over object-based features, to different forms of experience-dependent specialization. Previously, Testolin et al. (2017) investigated one mechanism of neuronal recycling–learning specialized letter features which take general low-level features as input. This partially specialized feature space showed several properties of human perception

including correlating with the perceptual similarity of letters, supporting letter decoding in noise-degraded images, and exhibiting superior letter decoding for fonts with lower perimetric complexity. These findings led Testolin et al. (2017) to conclude that letter perception primarily relies on domain-general visual features with some degree of domain-specific tuning. Our findings also support the claim that letter perception relies on domain-general features; however, by studying a broader range of neuronal recycling mechanisms we argue for a fully domain-general account of recycling. By investigating a hierarchy of general object-based features we found that mid-to-high level features exceeded low-level features in their ability to account for letter perception. These complex object-based features accounted for letter perception so well that a variety of models with partial letter specialization, including the model from Testolin et al. (2017), failed to show a superior match to human perception. Thus, the range of models we tested provide evidence that visual letter perception primarily relies on the recycling of features unaltered by letter learning.

Recent findings using other methodologies also corroborate our computational-behavioral evidence for this claim. For example, both readers and non-readers of an alphabet have nearly identical representational geometries for letters, as measured using a visual search paradigm over two different Brahmic scripts [44]. Further, in the macaque visual system, letters and words can be linearly decoded from neural responses of inferotemporal cortex in monkeys who have never been trained to distinguish between letters [59]. Together with our findings, these studies indicate that prior to learning to read an alphabet, the visual system already has complex general features which adequately distinguish between letters, and that learning to recognize letters may make at most negligible changes to this feature space.

An additional insight offered by this work is related to the complexity of the feature spaces. By leveraging the layer hierarchy of deep convolutional neural networks, we found that mid-to-late layers of the object-trained network, rather than the earliest layers, best accounted for the behaviorally measured similarity of letters. Thus, both qualitative characterizations of deep neural network feature tuning and intuitions about feature hierarchies suggest that the features underlying letter perception likely exhibit greater complexity than simple edge detectors [60]. The intermediate-level complexity of these underlying features clarifies why simple feature models failed to fully capture letter similarity structure [1,28–30] and why early data-driven attempts to infer the feature space resisted easy interpretation [1,20,27]. These layer-wise results also complement recent findings in macaques, in which letter identity could be more accurately decoded from inferotemporal cortex than from the earlier stage region of V4 [59]. It has been theorized that object contour representations, specifically line junctions, are recycled to represent letters [61,62], which may lead some to assume that low-level edge detectors should account for letter perception. However, studies of the perception of line drawings [63] indicate that object contours are actually fairly high-level representations abstracted from illusory edges created by illumination and shadows. Thus, while letters may appear to be relatively simple visual stimuli, our findings demonstrate that mid-to-high level features underlie our perception of letters, complementing previous research.

While our work here makes inferences about the nature of visual system representations through a behavioral-computational approach, our findings also lead to predictions about the representations of the visual word form area in the ventral stream. Neuropsychological studies have found that damage to the left occipitotemporal cortex (in the vicinity of the visual word form area) leads to slow and laborious letter-by-letter reading of words, though debates surround the extent to which such patients also show deficits in object perception [64,65]. In addition, studies on the connectivity constraints between language regions and occipitotemporal cortex support the view that information on letter strings is preferentially read-out from the visual word form area [66–70]. These neuropsychology and connectivity studies help

arbitrate on whether causally relevant information on letter strings is represented in a local region of visual cortex. However, these findings do not directly speak to the nature of the feature tuning in this region—whether the features are specifically tuned to distinguish between letters, or if they are more generally tuned to distinguish among all visual inputs. Based on our behavioral findings, we make the following brain predictions about responses to letters in the visual word form area. We predict that general object-based features will account for the representational space for letters both before and after letter learning. The pattern of response elicited by each letter in the visual word form area can be expressed as a vector with a certain direction and magnitude. We predict that letter learning increases the magnitude of each of these vectors without changing their directions. Such a change would preserve the relative similarity of each pair of letters, while increasing the distance between all letter pairs by a proportional amount, and thus might facilitate read-out of letter information.

Future studies are required to determine the extent to which our findings generalize to the context of reading. In this study, we chose to focus on individually presented lowercase letters in the Roman alphabet, raising the question of whether our claims would differ if our behavioral tasks and models were aimed at letter-string and word-level representations. This would be the case if the visual system learns specialized features for detecting combinations of letters en route to whole word representations [71]. Indeed, classic studies on the word superiority effect show that letter recognition is improved when letters appear in the context of a word [72,73]. However, focusing on individual letter-representations is not wholly unjustified, as there is some empirical support that letter string representations are primarily linear combinations of letter representations [44]. For example, the perceptual similarity of bigrams as measured through visual search is linearly predictable from the perceptual similarity of their constituent letters [44]. In addition, our work does not address the perception of non-alphabetical writing systems. Future behavioral-modelling work could leverage the approaches used here to investigate the extent to which other written symbol representations rely on general versus specialized feature spaces.

While the object-based spaces considered in our study account well for the perceptual similarity of letters, they did not reach the noise ceiling of our data, raising the question of what model could fully account for the behavioral data. General object-features and specialized letter features were operationalized in this paper through one class of models (discriminative, deep neural networks). Perhaps a different class of computational model could fully explain the perception of letters, for example, generative Bayesian models trained on the motor routines used to produce letters [74]. However, it is also possible that this small predictive gap for letter perception is simply due to the fact that object-trained neural networks only capture some aspects of human object perception [55,56,75–77]. For example, object-trained neural networks tend to represent local contour and texture features more so than global shape features [55,56]. Given how well the object-trained model in our study accounted for the perceptual similarity of letters, it is plausible that local shape features play a substantial role in the human perception of letters. However, if human letter perception also depends on global shape features, then this may explain why our object-trained network did not reach the noise ceiling of the behavioral data. As a first step in exploring this possibility we considered the representations learned from stylized-ImageNet [56], which relies less on local features in its classifications, though this network and the typical object-trained network did not consistently differ in how well they matched the perceptual similarity of letters. Perhaps the further discovery of object-trained models with more humanlike mixtures of local and global shape features would improve our ability to account for human letter perception. Another possibility is that the object-trained networks included in our study did not reach the noise ceiling of the data because they may not have been adequately trained to learn humanlike face representations,

given the relationship between cortical territories selective for words and faces [67]. Exploring all these possible alternatives is beyond the scope of this paper, thus this work leaves a small but theoretically important puzzle in accounting for the perceptual structure of letters.

While not the primary focus of this study, this work also corroborates prior findings on perceptual bottlenecks to cognitive tasks. Previous studies have found that measurements of perceptual similarity from a variety of visual tasks (e.g., tasks relying on working memory, visual search, categorization, and visual awareness) reflect the representational structure of occipitotemporal cortex [46,78,79]. In this study, the visual search experiment and categorization experiment involved different task demands and cognitive operations, but the representational structures measured in the two tasks were highly similar. These findings indicate that the representational similarity of a visual feature space creates a bottleneck for a number of perceptual tasks–for example, limiting working memory capacity as well as reaction times in visual search and categorization tasks. However, there are minor differences in the representational structures measured in each of these tasks, and future work could account for these differences by more explicitly modelling the cognitive operations which read out information from visual feature spaces.

Broadly, we view our investigation into letter perception as one case study within wider debates concerning the extent to which perception includes domain-general versus domain-specific mechanisms. We join other researchers operationalizing feature spaces with neural networks to gain insight into this debate [80]. These include investigations into the perception of a wide variety of domains of sensory input including faces [54,81]; objects, scenes, and near-scale "reachspaces" [82]; approximate number [83]; and language and music [84]. We see the approach taken in this study as broadly useful for determining the degree to which different domains of stimuli are represented by specialized mechanisms versus domain-general mechanisms.

## Methods

### Ethics statement

The experiments were approved by the Institutional Review Board at Harvard University. All participants gave informed written consent to participate in the study.

### Experiment 1: Odd-One-Out visual search

*Participants*. 222 participants completed the visual search experiment on Amazon Mechanical Turk.

*Stimuli*. The stimulus set consisted of 520 images of the 26 letters across 20 fonts. Each image depicted an individual letter, presented in lower case. The following fonts were used: Al Bayan, American Typewriter, Arial Black, Arial, Ayuthaya, Baskerville, Chalkboard, Comic Sans MS, Courier, Didot, Futura, Georgia, Helvetica, Hiragino Sans, Impact, Kefa, Kokonor, Luminari, Noteworthy, and Papyrus. All stimuli can be viewed in S7 Fig.

*Procedure*. Participants completed a visual search paradigm in which they viewed a set of stimuli and detected the odd one out as quickly as possible. On each trial, the target was one letter image, and the distractors were another letter image in the same font. Each trial started with the presentation of a blank circular arena with a fixation cross at the center for 1s. Next, six stimuli were presented in a ring around the fixation cross. The target stimulus was presented in one randomly selected position, while the distractor stimuli were presented in the other five positions. Participants were instructed to press the space bar as soon as they detected the location of the odd-letter-out. Immediately following this keypress, the positions were masked with a white-noise image for 250ms. Next, the numbers 1–6 displayed over the

positions, and participants reported the location of the target by pressing the corresponding number key. After the response, there was a 500ms pause, and the next trial automatically began.

Each Human Intelligence Task (HIT) completed by participants included 325 trials, one for each possible letter pairing. The target appeared at a random location on each trial and was equally likely to appear at any of the six locations. For each pair of letters, there were 6 possible target positions, 20 possible fonts, and 2 target-distractor assignments (e.g., a among b, or b among a), yielding 240 different combinations. Each of these combinations was tested in a separate HIT; therefore, target position, font, and target-distractor assignment were counterbalanced across 240 unique HITs. Because each HIT included all possible letter pairings, this design ensured that slower or faster participants equally influenced all letter pairings in the resulting representational dissimilarity matrix.

Five of the 222 participants completed multiple assignments (4–6 HITs). Due to an error in posting the experiment to Amazon Mechanical Turk, four participants completed the same assignment as others, so their data was excluded.

*Constructing the representational dissimilarity matrix.* The visual search reaction times were used to construct a representational dissimilarity matrix between pairs of letters. First, all incorrect trials were excluded (2.5% of trials). Then the reaction times from each HIT were log-transformed to account for the positive skew of the data (Palmer et al., 2011) and z-scored. Because each HIT tested all the letter pairs, these z-scores reflect the relative speed of each letter pairing in the distribution of all possible letter pairings. Trials that were more than three standard deviations from the mean were excluded. Next, we calculated the mean z-scored reaction time for each pair of letters across all the HITs. These values were multiplied by -1, so that higher numbers reflect more dissimilar letter pairs, and were used to fill the lower triangle of a 26x26 representational dissimilarity matrix (RDM).

*Noise Ceiling Calculation.* We determined the noise ceiling of the visual search data by measuring its reliability across HITs. We split the HITs in half by font, with each set containing all the HITs from ten of the twenty fonts. We calculated the RDM for each set of HITs, then we found the Spearman correlation between the two RDMs. This was conducted for all possible splits of the twenty fonts (184,756 splits). The mean of all these correlations was a measure of the reliability of a sample half the size of our full sample. Finally, we used the Spearman-Brown prediction formula ($\frac{2*\rho_{12}}{1+\rho_{12}}$, where $\rho_{12}$ is the reliability of half the sample) to estimate of the reliability of our full sample. We assume that this is the best we could expect any model to correlate with the data.

*Error RDMs.* While errors were not the planned target of the experiment, we also constructed RDMs based on the mean accuracy of each letter pair. The accuracy-based RDM was moderately reliable ($\rho = 0.44$) and was moderately correlated with the RT-based RDM ($\rho = 0.47$).

## Experiment 2: Letter categorization

*Participants.* 518 participants completed the letter categorization experiment on Amazon Mechanical Turk.

*Stimuli.* Stimuli were the same as the visual search experiment. Images were sized and positioned to align the baseline of the letters.

*Procedure.* Participants were shown single letters one after another and categorized them as quickly as possible. Each HIT consisted of two blocks of trials. At the start of each block, participants were given a target letter (e.g., "a") via visual text in font Arial. Each trial began with the presentation of an empty square arena for 750ms, after which a single letter in a random font

appeared in the middle of the arena. Participants responded as quickly as possible whether it was an example of the target letter or not. Participants pressed "["if the stimulus was an example of the target letter or "]" if the stimulus was not an example of the target letter. The next trial automatically began after the participant's response. If participants did not respond within 1250ms, then a screen appeared with the message: "Sorry, too slow! Answer quickly and correctly." If participants answered incorrectly, then a screen appeared with the message: "Try to answer each trial correctly."

The key measure we were interested in is the time it takes a participant to reject letters that did not match the target. We assume that letters that are more similar to the target will take longer to reject. For example, if the target for a given block of trials was "a" and the letter presented on a given trial was "b", then we would consider the reaction time of that trial to reflect the perceptual similarity between "a" and "b". Thus, we considered the reaction time of non-matching trials as a measure of the perceptual similarity of letters.

We measured the reaction times of all 650 possible pairs of targets and non-target letters across 664 HITs. To keep the length of each HIT under approximately 20 minutes while also obtaining reliable data, only a subset of all letter pairs could be measured in each HIT. Each HIT measured the perceptual similarity of 26 letter pairs, with 13 letter pairs included in each of the two blocks. Different target letters were assigned in the two blocks. Half the trials matched the target and half did not to prevent the creation of a bias toward either response. Each HIT included 416 trials, with each non-target letter repeated in 8 trials. Across all the HITs, each letter pairing was measured in 50–56 HITs. These experimental design choices were made with the goal of using linear-mixed effect modeling to estimate the reaction time of each pair of letters while accounting for differences in overall reaction time between participants. Before running the main large-scale experiment, we first ran a separate pilot study measuring the similarity of 50 letter pairs to ensure that our design and counterbalancing procedures would yield reliable data.

69 of the 517 participants completed multiple HITs, each completing up to 12 HITs. Data from 55 HITs were excluded because the trial accuracy was below 90%. Data from one HIT was excluded because it did not save properly.

*Constructing the representational dissimilarity matrix.* First, incorrect trials were excluded from each HIT. Next, the mean reaction time was calculated for each of the 26 target/non-target letter pairs tested in each HIT. If more than 4/8 trials for that pair of letters were incorrect, then the mean reaction time for that pair of letters was excluded. Across all the HITs, this data was compiled into a matrix with columns for three variables: 1) a categorical variable (Condition) specifying the letter pair tested, 2) a continuous variable (RT) specifying the mean reaction time for that letter pair, 3) a categorical variable (HIT) specifying the HIT in which that mean reaction time was collected.

We computed a linear mixed effects model to estimate the reaction time for each letter pair across the HITs with the following equation: RT~Condition+(1|HIT). The categorical variable of condition was dummy coded, so the model yielded an estimate of how each letter pair's reaction time differed from a reference letter pair. We added the estimated reaction time of this reference letter pair to every condition to get the estimate of their reaction times. For each pair of letters, there were two conditions corresponding to the two target/non-target assignments (e.g., the target is "a" when "b" is presented versus the target is "b" when "a" is presented). We took the mean of these two conditions to estimate the perceptual similarity of each pair of letters. The estimates of letter similarity were then multiplied by -1 to convert them into estimates of letter dissimilarity and inputted into the lower triangle of a 26x26 RDM.

*Noise Ceiling.* We determined the noise ceiling of the categorization data by measuring its reliability across measurements of each letter pair. Each letter pair was measured in at least 41

HITs after exclusion. We divided these measurements in half, so each split of the data included an equal number of measurements per letter pair. We calculated the RDM for each split of the data using the same procedure as above. Then we found the Spearman correlation between the two RDMs to measure the reliability of a sample half the size of our full sample. We completed this procedure across 10,000 random splits and calculated the mean split-half correlation. Finally, we used the Spearman-Brown prediction formula ($\frac{2*\rho_{12}}{1+\rho_{12}}$, where $\rho_{12}$ is the reliability of half the sample) to estimate the reliability of our full sample. We assume that this is the best we could expect any model to correlate with the data.

*Error RDMs*. While errors were not the planned target of the experiment, we also constructed RDMs based on the mean accuracy of each letter pair. The accuracy-based RDM was moderately reliable ($\rho = 0.52$) and was moderately correlated with the RT-based RDM ($\rho = 0.55$).

## Neural network models

**Primary models.** To obtain a model of general object-based features, we used an instantiation of AlexNet trained to do 1000-way object classification on the ImageNet database [31,50], available through the PyTorch ModelZoo. More specifically, AlexNet was trained on the subset of ImageNet used in the 2012 ImageNet Large Scale Visual Recognition Challenge [85].

To obtain a model of specialized letter features, we trained a randomly initialized instantiation of AlexNet to do 26-way letter classification on a database including typeset letter images we call GoogleFonts and handwritten letters from the NIST database [86]. By solely training this network with images of letters it could only learn features present in letter images (rather than also training it with object images, which would give the network the chance to learn both object and letter features). The GoogleFonts database includes 60958 images, with each lowercase letter depicted in 2344 different fonts. When creating this database, we included fonts which had all twenty-six lowercase letters, were not in cursive, and did not produce identical images to another font. Example images from the GoogleFonts database can be viewed in S7 Fig. To introduce more variability to the images our letter-trained models were exposed to, we included 1000 images per letter category from the EMNIST database [86] in the data set, bringing the total number of images per class to 3344. 20% of these images were reserved for a test set, while the other 80% were used for training.

Additionally, the following data augmentations were included during the training of the letter model: 1. Varying the size of the letter within the image such that the maximum size was 4x the minimum size in each dimension, 2. Varying the position of the letter to be uniformly distributed in the image plane, 3. Uniformly distributed random tilt between ±15 degrees, 4. Uniformly distributed random horizontal and vertical shears with a factor between .8 and 1.25. 5. Random letter and background color, under a minimum contrast constraint. 6. Gaussian additive pixel noise with a standard deviation uniformly selected from .01-.1 per image (given black/white pixel values have been normalized to 0/1). We applied each of the above augments with a 60% probability during training to ensure that the model was exposed to typical letter images in addition to the augmented images. This allowed for the models to learn to classify both augmented letters and the more typical letters used in the human experiments. Example augmented images can be seen in S8 Fig.

The model was trained on minibatches of 64 images for 100 epochs using the Adam optimizer [87], with betas .9 and .999, and an initial learning rate of .001. The model epoch with highest test set accuracy was selected for comparison with human subjects.

Starting from different random seeds, we trained six iterations of AlexNet on letter classification. Model-behavior correlations showed a similar trajectory across all model iterations. To

pick the model iteration with the most typical representational structure, we used the following procedure: First, we computed the rank-order RDMs for each layer of each network. Second, we computed the mean layer-wise RDMs across the network iterations. Third, for each network iteration we computed the Spearman correlation between its layer-wise RDMs and the mean layer-wise RDMs. Finally, we picked the model with the highest correlations between its layer-wise RDMs and the mean layer-wise RDMs.

Additionally, reasoning that AlexNet has far more parameters than necessary for letter classification, we created another model of specialized letter features by training a smaller five-layer CNN on the GoogleFonts and EMNIST database in the same manner. Architecturally, this network had 3 convolutional layers followed by 3 fully connected layers, with 4x4 adaptive average pooling in between. The first convolutional layer had 5x5 kernels, with a stride of 2, padding of 3, and 40 output channels. The next two convolutional layers had 3x3 kernels, with strides and padding of 1, with 20 and 60 output channels respectively. The fully connected layers had 500, 200, and 26 output channels, with 50% dropout between layers during training. All layers used ReLU activations.

We also trained an AlexNet model to classify letters on scene backgrounds. To train this network we collected a set of 7,071 scene images from the SUN database. Scene images were excluded which had clear text. These scene images were randomly included as backgrounds for 100% of the images during training. The same set of augmentations were used as before (except the color background variation was replaced with scene backgrounds).

All model test set accuracies can be viewed in S1 Table.

**Models of object-based features with experience-dependent specialization.** Two different fine-tuned networks were created, both starting with an AlexNet pre-trained on ImageNet. For the first fine-tuned network, we added 26 randomly initialized dimensions to the final classification layer to create a 1026-way classifier, then trained the network on both ImageNet and the letter database. The network was trained on minibatches of 128 images, each with a random collection of Imagenet and letter images. As there were many more Imagenet training images in the dataset, each letter training image was shown 3 times to the network per epoch. The model was trained for 30 epochs with an initial learning rate of .0001. For the second fine-tuned network, we replaced the final 1000-way classification layer with a randomly initialized 26-way classifier. This network was then fine-tuned only on letters for 100 epochs, and the model epoch with highest test set accuracy was selected for comparison with human subjects. Letter data was augmented during training as described above for both these models.

To create specialized letter networks branching off ImageNet-trained AlexNet we took inspiration from the methods introduced by Kell et al. (2018). Five different models were created, each with a different branching point off the base AlexNet at one of the first five ReLU layers. Each network branch was architecturally identical, varying only in the number of input channels in the first layer, in order to match the number output channels from the different AlexNet layers. Besides this difference, each branch is architecturally identical to the custom small CNN architecture described above. Note that the weights of pre-trained base AlexNet were frozen, so feature learning only occurred in the network branches. The branching networks were trained on the letter database in the same manner as letter-trained AlexNet.

Finally, we used the following procedure to identify any letter-preferring features found across the layers of both the primary AlexNet trained on ImageNet model, and of the first fine-tuned AlexNet model (fine-tuned on a mixture of ImageNet and letters). First, we measured activations to the 50,000 image ImageNet test set and the 2860 image GoogleFonts test set, evenly sampled by category. Next, for each feature in each network, we conducted a two-sample t-test comparing the activations to the 26 letter categories of GoogleFonts and the 1000 object categories of ImageNet. Finally, features were selected which exhibited greater activation to letters and a p-value of $< 0.05$.

To select features which did not prefer letters, we identified features which responded to any letters, and which were not previously identified as letter-preferring. For each layer of each network, we selected 100 random samples of these features, matching in size to the letter-preferring features of that layer. When comparing the letter-preferring and non-letter-preferring subspaces, the mean model-behavior correlation was computed across the 100 samples of non-letter-preferring subspaces.

**Measuring activations and creating RDMs.** Feature activations to the 520 experimental stimuli were measured in each convolutional neural network, from the ReLU stages of each layer. For convolutional layers, we computed the total amount each image activated each feature by summing the activation maps of each feature. For example, Layer 5 of AlexNet has 256 features each with an activation map of dimensionality 13x13; by summing across the activation maps we obtained a 256-dimensional vector for each image. This step was taken to make the RDMs measured from our models comparable to the RDMs measured in our behavioral tasks. In visual search, participants compare the features of multiple stimuli at different visual locations, requiring abstracting from each stimulus's retinotopic location. One way this comparison could be accomplished is by computing the total amount each feature is activated by each stimulus, then comparing the feature activations between stimuli. We chose to model such a comparison mechanism here.

After computing feature activations to each stimulus, representational dissimilarity matrices were then computed for each layer of each network. The procedure for making these RDMs differed between the two experiments to best parallel the tasks completed by participants. For Experiment 1, participants only ever directly discriminated between letters of the same font, so the model RDMs for this experiment only included dissimilarities between letters of the same font. Specifically, activations were measured for each letter stimulus, and the Euclidean distance was computed between all pairs of letters within each font, yielding a 26x26x20 matrix (26 letters x 26 letters x 20 fonts). The mean of this matrix was calculated across fonts to create a 26x26 RDM for each layer of each neural network. For Experiment 2, participants categorized letters across random font assignments, comparing letters in a specific font to a target letter, so the model RDMs for this experiment included dissimilarities between letters of all fonts. Specifically, the mean activations were calculated for each letter across font, providing an estimate of the target letter in each feature space. The Euclidean distance was computed between the activations of each font-specific letter and the mean activations of each letter. Then the mean of these distances was computed for each letter to create a 26x26 RDM for each layer of each neural network. To ensure that our results were not dependent on our choice of distance measure, we also conducted all analyses after computing RDMs with correlation distance and cosine distance, and we found that all the patterns of results were consistent across distance metrics.

**Model of a purely categorical representational space.** We created a model RDM for a purely categorical representational space, then computed the correlation between this model RDM and the RDMs from object-trained and letter-trained AlexNet. This model RDM was created from a 26-D representational space in which each dimension represents the presence of each letter identity with a binary 0 or 1. The 520 experimental images (26 letter identities across 20 fonts) were used to create a 520x520 RDM.

**Intuitive feature model.** We also computed a model RDM using a set of intuitive features previously described in the literature on letter perception (Fiset, 2008; Wiley, 2016; Wiley, 2020). The features were the following: straight lines at different orientations (vertical, horizontal, slanted right, and slanted left), curved lines (open on the right, left, bottom, and top), intersections (two-, three-, and four-way), line terminations (on the right, left, bottom, and top), diacritics, symmetry, and closed space. We judged the number of times each feature was

present in each letter image used in our experiments (author DJ completed the ratings). As above, Euclidean distance in the feature space was used to construct the model RDM.

**Computing model-behavior correlations.** For each experiment, we compared the behaviorally measured RDM with each of the RDMs from our convolutional neural networks to determine which model features best matched human behavior. We took the values below the diagonal of each RDM, then computed the Spearman correlation between the behaviorally measured RDM and each of the model RDMs. See S1 Text for further explanation of the theoretical assumptions underlying the way we relate the behaviorally measured RDMs and model RDMs. Note that we did not conduct RSA-reweighting when computing model-behavior correlations. This sets a higher bar for models to exhibit high model-behavior correlations. Network feature spaces could fail to capture the perceptual similarity of letters because they have the wrong features or features in the wrong proportions.

**Comparing model-behavior correlations between networks.** We conducted bootstrapping statistical tests to compare pairs of networks to see which yielded the highest correlation to the behaviorally measured similarity of letters. Our experiments measured the similarity of 325 letter pairs, so we bootstrapped 50,000 samples of 325 letter pairs. We compared pairs of networks two ways: i) by their maximum model-behavior correlations, and ii) layer by layer. To compare networks by their maximum model-behavior correlation, we selected the layer from each network which exhibited the highest model-behavior correlation in the original sample, then we determined which of the two feature spaces had the higher model-behavior correlation for each bootstrapped sample. To compare pairs of networks layer by layer, we compared the model-behavior correlations for each layer for each bootstrapped sample.

## Supporting information

**S1 Text. Supplementary information.** Additional details provided for the decoding analyses from object features, model comparisons, and how we related behavioral measurements to neural network feature spaces
(PDF)

**S1 Fig. Linear decoding of letter identity from object-trained features.** A. Linear support vector machines were trained to categorize letters across font and size variation from the features of AlexNet trained on ImageNet. Three subsets of features are compared: 1) all the features from each layer, 2) only the features from each layer which preferentially responded to letters over object images, 3) a random subset of features matching the number of letter-preferring features. Classifiers were trained on random sets of letter fonts and sizes, then tested on left out fonts and sizes. The shaded areas indicate the 95% confidence interval across random testing/training splits. For the random subset of features, the confidence interval also includes variance introduced by the random selection of features during each instance of classifier training and testing.
(TIFF)

**S2 Fig. MDS plot visualizations of letter similarity as measured in behavior and the two primary feature models.** Multidimensional scaling was used to project 26x26 RDMs onto two dimensions. Distance between letters illustrates their similarity as measured during visual search and letter categorization (above). Layer-wise MDS plots for object-trained AlexNet (middle) and letter-trained AlexNet (bottom) are also illustrated. Please note that reducing the dimensionality of neural network feature spaces to two dimensions obscures a lot of meaningful variance, and these visualizations are only for exploratory inspection.
(TIFF)

**S3 Fig. Model-Behavior Correlations for Letter Features from Testolin et al. (2017).**
Model-behavior correlations are plotted on the y-axis, as a function of the layer of AlexNet trained on ImageNet. Model-behavior correlations for the letter-trained features from Testolin et al. (2017) are plotted in orange. The shaded error range indicate the 95% confidence interval across bootstrapped samples of letter pairs.
(TIFF)

**S4 Fig. Comparisons of object-trained and letter-trained networks with AlexNet with random weights.**
(TIFF)

**S5 Fig. Model-Behavior Correlations for a Smaller Network Trained on Letters Categorization.** A smaller architecture (see Methods) was trained on 26-way letter classification to create another model of specialized letter features. Model-behavior correlations are plotted on the y-axis, as a function of the model layer. The shaded error range indicate the 95% confidence interval across bootstrapped samples of letter pairs.
(TIFF)

**S6 Fig. Model-Behavior Correlations When Letter-trained Models Have Less Varied Input.**
The letter-trained model shown here was trained on an image set with 550 typeset fonts per letter with size augmentation. In comparison, the letter-trained models in the main text were trained with 3344 typeset and handwritten letters across augmentations of size, position, skew, tilt, color, and noise.
(TIFF)

**S7 Fig. Example letter images.** A. Images used in the two behavioral experiments: all twenty-six lower case letters across twenty fonts. B. Example images from the GoogleFonts database used to train specialized letter networks. The full database includes all twenty-six lowercase letters across 2344 fonts.
(TIFF)

**S8 Fig. Example augmented letter images used when training letter-classifying neural networks.** Augmentations included size, position, tilt, shearing, Gaussian noise, and color. For a full description of augmentations see the Methods section.
(TIFF)

**S1 Table. Summary of neural network models.** Here we list the model-behavior correlations (Spearman rho) of each neural network model. In addition, we list their accuracies on ImageNet or letters depending on which database they were trained on.
(TIF)

## Author Contributions

**Conceptualization:** Daniel Janini, Talia Konkle.

**Data curation:** Daniel Janini.

**Formal analysis:** Daniel Janini, Chris Hamblin.

**Funding acquisition:** Daniel Janini, Talia Konkle.

**Investigation:** Daniel Janini, Talia Konkle.

**Methodology:** Daniel Janini, Chris Hamblin, Arturo Deza.

**Project administration:** Daniel Janini.

**Resources:** Daniel Janini, Talia Konkle.

**Software:** Chris Hamblin, Arturo Deza.

**Supervision:** Talia Konkle.

**Visualization:** Daniel Janini, Talia Konkle.

**Writing – original draft:** Daniel Janini, Talia Konkle.

**Writing – review & editing:** Daniel Janini, Talia Konkle.

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
