## [Decision Letter · Decision Letter 0]

6 Jan 2022

Dear Mr. Janini,

Thank you very much for submitting your manuscript "General object-based features account for letter perception better than specialized letter features" for consideration at PLOS Computational Biology.

As with all papers reviewed by the journal, your manuscript was reviewed by members of the editorial board and by several independent reviewers. In light of the reviews (below this email), we would like to invite the resubmission of a significantly-revised version that takes into account the reviewers' comments.

The reviewers all agreed this paper addresses an important question, and had largely positive feedback about the methods and paper. There were however major concerns, particularly about the format of the behavioral task, the training of the letter network and its comparison to the object-trained network.

We cannot make any decision about publication until we have seen the revised manuscript and your response to the reviewers' comments. Your revised manuscript is also likely to be sent to reviewers for further evaluation.

Sincerely,

Leyla Isik

Associate Editor

PLOS Computational Biology

Thomas Serre

Deputy Editor

PLOS Computational Biology

The reviewers all agreed this paper addresses an important question, and had largely positive feedback about the methods and paper. There were however major concerns, particularly about the format of the behavioral task, the training of the letter network and its comparison to the object-trained network.

Reviewer's Responses to Questions

**Comments to the Authors:**

Reviewer #1: In this paper, “General object-based features account for letter perception better than specialized letter features”, the authors tackle a timely and interesting topic in an interdisciplinary fashion. They perform large-scale behavioural experiments on letter perception and subsequently use normative modelling (i.e. task-trained convolutional neural network models) to better understand the patterns observed. They conclude that letter perception remains best explained by category-trained DNN feature spaces, providing an interesting perspective on neuronal recycling, and visual plasticity during reading acquisition. The paper is very clearly structured and was a pleasure to read. The figures are highly informative, and the available code will aid reproducibility. In short, I very much enjoyed this paper. Yet, I have a few concerns which I hope the authors will address in the resubmission to solidify the main claims. I very much hope that my comments will be perceived as constructive as they are intended and that they may help the authors improve their manuscript going forward. I look forward to our discussion.

Signed

Tim Kietzmann

Major:

1. Both behavioural tasks, including the categorization task, seem rather perceptual in nature. This view is backed up by the high inter-experiment correlation reported in lines 290ff. As a result, similar letters are “close to each other” in the behavioural data. In contrast to this, the letter-trained DNNs are optimised for letter discrimination. As a result, perceptually similar letters will be projected onto feature axes that ideally distinguish between them. Given this observation, I am unsure in how far we should expect letter training to successfully capture data that relies on perceptual comparisons.

2. The behavioural tasks and DNN analyses operate on rather different levels. Whereas the human data is collected via RTs, the networks internal representations are probed without a behavioural focus. Would it make sense for the authors to consider behaviour for both, models and brains? One way of doing this would be to train linear readouts on the various models in the tasks that humans engaged in, and to take the “distance to hyperplane” as a proxy for RTs.

3. I am unsure how much we can trust the results of training AlexNet on comparably few letter data. AlexNet has ~61M parameters, but letter training was performed using only 14k datapoints. As a result, the model features will heavily overfit to the training data, likely amplifying my concern #1 above. In contrast to this, ILSVRC trained models have seen 1.3M images (~100x more data, not even considering the fact that the convolutional kernels benefit largely from input to all units in natural stimuli vs very sparse input for letters). Given this stark difference in the number of training samples, it is perhaps less surprising that model fine-tuning with 1026 instead of 1000 image categories led to comparably small changes in the model RDMs. In short, I feel like the models behave as they should. ILSVRC training will be quite similar to ILSVRC+letters, and training on letters only will lead to a model that can easily distinguish between them, thereby leading to a worse fit to tasks that rely on perceptual similarity.

Minor:

1. Very small detail, but it could be clarified that AlexNet was trained on ILSVRC 2012, which represents only a small fraction of ImageNet.

2. Could I ask the authors to clarify how exactly the model RDMs were computed? The methods section states that “we computed the summed activation map of each feature for each stimulus”, but it remains unclear to me what actually happened. The more traditional approach would be to take each network unit as its own dimension, but this seems to not have happened here. Moreover, could the authors comment on whether RSA-reweighting (e.g. Storrs et al.) would make a difference to the results obtained?

Reviewer #2: In manuscript reports a study investigating hypotheses about whether letter representation depends on learning to recognize special letter features or by reusing general visual features acquired through object recognition. Similarity judgments for letters (visual search and visual categorization) were compared to the layer representations of neural networks either trained to do standard 1000-class image classification or 26-class image classification. Using representational similarity analysis, it was found that the later layers of the object-trained networks showed better correspondence to the behavioral data for both tasks. The authors interpret these results in favor of the domain-general learning and reuse hypothesis.

I think this is a topic in which NNs do provide a useful tool for evaluating different hypotheses. Overall, I think it is a great project to be pursuing. I have some serious concerns though about how the results are interpreted and whether enough has been done to really show that an object-trained network is superior to one specialized for representing letters. As outlined in some of my major comments, I think a lot more work would need to go into how the letter network is trained in order to make a reasonable comparison to the object trained network. Along these lines, below are several comments/suggestions about analyses that could be carried out (some simple, some rather laborious). I hope the authors find them to be constructive. Ultimately, I think the conclusions that can be drawn from this study are, at present, rather limited as are the findings.

Major Comments:

A. The authors say (L132-133):

“These results provide clear empirical support for the theoretical position that letter perception is primarily supported by general object-based features, rather than specialized features.”

I do not see how these sorts of results could do this, even in principle. Taken at face value, the results show that the object recognition trained network better captures the letter similarity judgments and so it is *possible* to have an architecture that might represent the similarity of letters without being trained to represent a letter feature space. Showing this possibility is an interesting, if modest, result. But showing such a possibility is not sufficient as evidence in favor of the psychological hypothesis that letter perception is “actually” generated by such a learning process. Especially not when an ImageNet trained network does not even obviously “see” global shape (see comment below), which is presumably required for representing letters under either of the hypotheses considered.

To be clear, I am extremely sympathetic to the reuse hypothesis that the authors think their results support. I just don’t think it gets support from these results. At most this is a kind of proof of principle for an aspect of the hypothesis.

B. There are a few analyses that I think would be useful for clarifying these results, which would not require much additional work by the authors.

First, how do the results look if one looks font-by-font, across the 20 different fonts? That is, makes a different 26 x 26 matrix for each of the 20 font, rather than averaging (L752-L753). This would be useful for showing the robustness of the results; that the performance is consistently higher across fonts.

Second, I think it would be useful to have a baseline model of the stimulus properties to compare to the behavioral judgments and neural data. For example, make 26 x 26 pixel similarity models of each font, and then average all 20 of these matrices to get the total pixel similarity model. How much would the observers performance be predicted by this model on the two tasks? Does it better capture the representation of the two NNs? This might also help clarify how they are each representing things.

Third, I think it would be helpful to have a baseline NN as well. For this, maybe see what results one gets with a completely untrained model, since even untrained DNNs have shown some correspondence to neural responses. Might help the authors case if an untrained model even does better than the letter trained model!

C. Beyond the issue raised in A, comparing the networks in this study really seems like apples and oranges. The nature of the two sets of training images are so radically different in their properties, in ways that we know matter to the training and generalization of NNs, that there could be many reasons for explaining the difference in performance that has little do with the hypotheses under consideration. Unless I missed it (and please, correct me), the only variation used was in the fonts used (550 total). But what about:

-Variation in the size, position, contrast, rotation in the picture plane, or even rotation in the depth plane. Addition of noise (cf. Testolin et al. 2017).

-The presentation of the characters in natural scene backgrounds in which they have been convolved but are still discriminable.

-What about training them to do classification of 26 letters x 20 behaviorally tested fonts. This would be more similar to what you have in ImageNet with different breeds of dogs, rather than all dogs being treated as alike. The 520 different categories would then at least give some semblance on similarity in scale of the classification task being performed.

-To really test this, you would need an ImageNet *of letters* in which letters appear in natural scenes with natural variation in viewpoint, illumination, partial occlusion, and so on and so forth. One might even just take pictures of pages of the sorts of books that we use to teach children letters. But I think there would have be some sort of matching of natural variation in the images to make an object trained network and a letter trained network comparable.

-One suggestion in line with the previous comment: do “stylized” letters like Geirhos et al. (2021) do for objects to improve representation of shape.

D. The authors cite studies that highlight the difference between how humans and DNNs represent objects (citations 67-71). Some of those studies though point to the ways in which DNNs do not really represent global shape (e.g. Baker et al. 2020, Geirhos et al. 2021). In order for the comparison of models to be most informative, I think one cannot be just relying on standard ImageNet trained networks unless it is just to make the “mere possibility” claim, per comment A above. I think one would have to do something to try and show that the networks are actually representing shape properties, since under anyone’s view, letters are represented as complex two-dimensional shapes on surfaces (or so I would hope).

For example, to add a suggestion to the list from comment C above, suppose one made a stylized-Letters image set a la Geirhos et al. 2021. Their results suggest that a network trained on stylized object images better represents the actual shape. In fact this is someone one could do for both the objects and letters, stylizing in similar ways. This would make the image sets more comparable, and if the object trained network was still better, then this would indeed support the possibility claim from comment A above.

Minor Comments:

-Capitalization for figure panels changes (L152 vs L156).

-L186 “First, visual feature spaces” ?

- If the similarity is going to be plotted for visual search (Figure 1B) it should be plotted for the categorization data in Figure 2.

-What are the between task reliabilities when split by font? I think it would be helpful to report this in the supplemental.

-I wonder if some of the visualization in Sup. Figure 3 could/should be in the main text.

-I might have missed it, but what were the layer-by-layer correlations between the object and letter networks?

-L473 Although it is tantalizing to know that there is an ongoing fMRI study on this, I think this sentence can be cut.

Reviewer #3: Janini and colleagues investigate the representational space of letter perception in humans using behavior and computational approaches. They find that a neural network trained on general object discrimination matches human behavioral performance on two letter discrimination tasks better than a neural network trained to discriminate letters specifically. The results provide strong support for a domain-general account to letter perception. These data provide important insights into how humans learn and perceive letters. This study will be of broad interest to the cognitive psychology and neuroscience community. Overall, this is a very nice study. Aspects of the analytic approach and data need clarification and further illustration. The authors should better address the degree of task dependency of their results. The discussion section would benefit from discussion of case studies and clarification on the broader impacts of the results.

Specific comments

The finding of correlations between behavior and the object-trained model close to the noise ceiling provides good support for the authors' domain-general account of cortical recycling in letter perception. In general, I agree with this interpretation, but the task demands are particularly important here. Would a task that is geared more towards word reading support a similar conclusion? This is particularly relevant when discussing the current results in relation to the VWFA. Also see the next comment re: behavior of individuals with alexia.

Overall, there is good discussion of how the results relate and add to prior literature. Additionally, I would like to see discussion of the relation to alexia arising from damage to the VWFA (e.g. Turkeltaub et al. 2014 Neurocase, also Behrmann et al. 1998 Neuropsychologia) that can result in letter-by-letter reading. These lesion cases are highly relevant for the discussion on domain-general vs -specific and (lack of) localization to a "letter subspace".

The decrease in model-behavior correlations at higher layers for the letter-trained network is striking. Presumably this network does just fine at discriminating letters. Some investigation into what's going on would help in interpreting comparisons with the object-trained network. e.g. looking at the MDS plots in Supplementary Figure 3, letters appear to be more dissimilar from one another in the letter-trained network. I wonder if there is an interesting story here about capacity limits to cortical feature spaces.

How does fine-tune training affect object-trained networks? How different are the RDMs after training? What was the accuracy for letter discrimination in these networks compared to the letter-trained network?

How (dis-)similar are letters across fonts (both for trained and tested)? I would expect changes in the representation space from learning to be strongest for letters that that have a good degree of variability across fonts. Given that much of the analyses focused between letter comparisons within a font or mean activations across fonts, would analyses targeted on variability across fonts reveal letter-specific effects of learning?

Relatedly, (as I understand it) the model RDMs for experiment 2 were computed by first calculating the mean activations for each letter across fonts. This doesn't appear well matched to the experimental design where subjects indicated whether single letters at specific fonts matched a target letter. Wouldn't it make more sense to calculate the distances between activations for font-specific individual letters compared to the target?

How do model-behavior correlations for the letter-preferring subspace compare with correlations for non-letter preferring subspaces (matched in # of units to letter-preferring)?

Overall, the discussion section is well written, but the claim of a fully domain-general account of letter perception (p 21) appears at odds (and undercut) by the later discussion of possible domain specialization (page 24). This comes across as trying to appeal to two (often conflicting perspectives) on cortical specialization and dampens the impact of the study.

The authors propose that letter perception is supported by fully domain-general representations whose features are unaltered by letter learning. Why is this still considered cortical "recycling"?

Minor comments

What was the behavioral accuracy for both experiments? Was accuracy and RT correlated across letter pairs?

For the neural networks trained on letters, what was the accuracy in discriminating letter pairs?

Please include the input layer in the model-behavior correlation plots.

I may have missed this in the methods - how were target letters given for experiment 2? Auditorily or visually? If visually, what font was used?

**Have the authors made all data and (if applicable) computational code underlying the findings in their manuscript fully available?**

Reviewer #1: Yes

Reviewer #2: **No: **Only the behavioral data and stimuli are up on OSF. They could have at least uploaded the network RDMs.

Reviewer #3: Yes

PLOS authors have the option to publish the peer review history of their article (what does this mean?). If published, this will include your full peer review and any attached files.

Reviewer #1: **Yes: **Tim C Kietzmann

Reviewer #2: No

Reviewer #3: No
---

## [Decision Letter · Decision Letter 1]

31 May 2022

Dear Mr. Janini,

Thank you very much for submitting your manuscript "General object-based features account for letter perception" for consideration at PLOS Computational Biology.

As with all papers reviewed by the journal, your manuscript was reviewed by members of the editorial board and by several independent reviewers. In light of the reviews (below this email), we would like to invite the resubmission of a significantly-revised version that takes into account the reviewers' comments.

In particular, Reviewers 2 and 3 raise concerns about the pattern of results (both the RDMs and comparison to the pixel model), and Reviewer 2 has further expanded on their suggestion to add natural background to the letter stimuli.

We cannot make any decision about publication until we have seen the revised manuscript and your response to the reviewers' comments. Your revised manuscript is also likely to be sent to reviewers for further evaluation.

Sincerely,

Leyla Isik

Associate Editor

PLOS Computational Biology

Thomas Serre

Deputy Editor

PLOS Computational Biology

Reviewer's Responses to Questions

**Comments to the Authors:**

Reviewer #1: The authors have done a great job at addressing my concerns, congratulations on a revision well done.

The paper can likely be published as is, but I would like to take this opportunity to, nevertheless, highlight one subtle remaining concern about the take home message provided, for example in the author summary. To me, the paper shows nicely that letter perception can be well explained by more general visual features useful for object recognition, rather than requiring extensive training on domain-specific letter data. This, however, does not directly speak to the question which features humans use to recognize/classify letters while reading. Put differently, classification of letters into different categories may rely on different mechanisms than judging their visual similarity (which could be the implicit task studied here given the experimental setup). Maybe this subtlety could be acknowledged more explicitly. A minor point, I hope the authors forgive my pedantry.

Reviewer #2: I want to thank the authors for their careful consideration of my comments on the previous version of the manuscript. They have carried out many of the analyzes I suggested, along with changes made based on feedback from the other reviewers. I am sure it was quite a bit of work to make these revisions. Overall, I think the manuscript is substantially improved.

I still have two remaining concerns, though I am confident that the authors can address them.

Major comments:

A. In the previous round I worried that the study involved comparing “apples to oranges” because the letter-trained network only sees letters varying in font. Based on my recommendation, they added many degrees of variation (size, color, noise, etc.) and still found the same results. That was great to see (both that the analyses were done and that the results were the same!). However, the real underlying concern was that the letters were presented without natural scene information. Here the authors made the case for not including this analysis, because perhaps the network would learn to recognize objects to boost recognizing letters, for example.

However, I am afraid I am not convinced, but perhaps I wasn’t clear enough about what I had in mind. Whether showing photos of letters on surfaces in natural images might create problems, most simply scenes could be included in a counterbalanced way as used extensively in the work out of James DiCarlo’s lab. In that group’s stimuli (which I am confident the authors are aware of), different rendered objects are presented on random scene backgrounds. There is no relationship between scene and object type, so (for example) a DNN trained on their images does not learn to recognize scenes in order to categorize objects. But it nonetheless must learn to recognize objects in the presence of scene information. Doing something like this for letters would suffice to address my apples-to-oranges concern. Even like the DiCarlo group, you could even show the scenes in grey scale, and randomly present them behind the letters. This would rule out that it is the presence of natural scene information that explaines the difference, and would strengthen the case that it is something about representing object properties that explains the superior correspondence for the object trained model.

B. Per my suggestion, the authors carried out further analyses looking at the correlation with a pixel model and an un-trained model. The pixel results are plotted in Figures 1 B-C, and show an equivalent effect for the object and letter trained models through the first two conv layers, rho = ~ 0.4. For the object trained network the correlations increase at the third layer and basically stay at the same level. In contrast, for the letter-trained model correlations rise for both tasks at layer 4, and then start to freefall reaching 0 for the fc layers for the visual search RDM and rho = ~ 0.2 for the letter categorization task.

It was not salient until having the pixel model as a benchmark, but it seems like something strange is going on here with the letter trained model. It would be one thing if the correlations remained above the pixel correlation level through layers 5-7, but were simply lower than the object trained network. But this seems to show that the fc layers have learned a representation that is almost entirely *unrelated* to the letter dissimilarities.

I am not sure what is going on with this. Minimally I think the authors need to discuss why this might be the case and the implication for interpreting the results (I looked for a place where this was already done in the manuscript. If I missed it, I apologize). For my part, it makes the story less about how much better the object model does and rather how poorly the letter model does, which is not quite as compelling. My instinct though is that something weird has happened with the training of the letter model, but I am not sure what.

Just a hunch, but I wonder whether this drop in correlation at final conv and fc layers would happen if the authors took up my suggestion from comment A and trained the letter networks with letters presented on random natural scene backgrounds. If the correlations stayed above the pixel level in that case for the letter trained model through the later layers, I think that would substantially strengthen the results. But even if there is still a drop, it would at least rule out the possibility that it has something to do with the model not being exposed to natural image properties.

I also wonder if the drop in correlations is somehow related to the issue raised by Reviewer 1 about the letter trained model being optimized for discriminating letters, not representing their similarity. But that is just speculation on my part.

Minor comment:

In the new added text (blue) I noticed p-values are sometimes reported without the effect size of the accompanying measure. I think it would be helpful to say what the rho value was in the text in those cases.

Reviewer #3: The authors have provided an extensive revision with new simulations, expanded stimulus sets for training, and important clarifications to the main text. Overall, I remain quite positive about the manuscript. These new analyses and revisions have strengthened the main claims, though also raise several new concerns that need to be resolved.

Looking at the RDMs in Supplementary Figure 2, the similarity structure for the updated letter-trained AlexNet looks odd in the deeper layers. It seems likely that the main reason for the poor match to behavior is due to the substantial distance of letters k and m from the rest of the letters. It is not clear why those letters would emerge to be so dissimilar in their activation patterns in comparison to all other letters. Perhaps this anomaly is masking a good correspondence with behavior for all other letters. This distinction of k and m from the rest of the letters was not present in the prior version of the model and needs to be resolved.

What was the motivation for setting the shallower network's depth to be layer 5? Looking at results from the AlexNet in the main analyses, model-behavior correlations were strongest at layer 4 then took a nosedive. From that, wouldn't a shallow network capped to the layer 4 make more sense (ie only layers that showed good correlation with behavior)? In the prior version of the manuscript, the shallow network was set to a depth of layer 4. It's unclear why the authors added an extra layer in the revision.

The authors make an interesting prediction that letter learning will not create new patterns of responses to letters in the VWFA, but rather will increase the amplitude of those patterns. I struggled with this distinction. Wouldn't increasing the amplitude of responses to letters change the patterns - i.e. result in a different similarity structure? This claim needs to be unpacked. It's unclear how this prediction would manifest in the models tested in the manuscript.

The additional model comparisons (eg pixelwise comparisons and stylized AlexNet) are all nice additions, but need better motivated in the results to explain how these serve as tests of robustness. For example, the use of a shallower network in the following paragraph was well motivated in the following paragraph.

**Have the authors made all data and (if applicable) computational code underlying the findings in their manuscript fully available?**

Reviewer #1: None

Reviewer #2: Yes

Reviewer #3: Yes

PLOS authors have the option to publish the peer review history of their article (what does this mean?). If published, this will include your full peer review and any attached files.

Reviewer #1: **Yes: **Tim C Kietzmann

Reviewer #2: No

Reviewer #3: No
---

## [Decision Letter · Decision Letter 2]

29 Aug 2022

Dear Mr. Janini,

We are pleased to inform you that your manuscript 'General object-based features account for letter perception' has been provisionally accepted for publication in PLOS Computational Biology.

Best regards,

Leyla Isik

Academic Editor

PLOS Computational Biology

Thomas Serre

Section Editor

PLOS Computational Biology

Reviewer's Responses to Questions

**Comments to the Authors:**

Reviewer #2: The further analyses and edits to the manuscript have addressed my two remaining concerns.

Reviewer #3: The authors have provided a thorough revision that addresses my remaining concerns and improves the clarity of the findings. This is a very nice paper and can be published as is.

**Have the authors made all data and (if applicable) computational code underlying the findings in their manuscript fully available?**

Reviewer #2: Yes

Reviewer #3: Yes

PLOS authors have the option to publish the peer review history of their article (what does this mean?). If published, this will include your full peer review and any attached files.

Reviewer #2: No

Reviewer #3: No

---

## [Editor Report · Acceptance letter]

21 Sep 2022

PCOMPBIOL-D-21-02106R2 

General object-based features account for letter perception

Dear Dr Janini,

I am pleased to inform you that your manuscript has been formally accepted for publication in PLOS Computational Biology. Your manuscript is now with our production department and you will be notified of the publication date in due course.

With kind regards,

Anita Estes
